# Repetitive Transcranial Magnetic Stimulation (rTMS) of Dorsolateral Prefrontal Cortex May Influence Semantic Fluency and Functional Connectivity in Fronto-Parietal Network in Mild Cognitive Impairment (MCI)

**DOI:** 10.3390/biomedicines10050994

**Published:** 2022-04-25

**Authors:** Sabrina Esposito, Francesca Trojsi, Giovanni Cirillo, Manuela de Stefano, Federica Di Nardo, Mattia Siciliano, Giuseppina Caiazzo, Domenico Ippolito, Dario Ricciardi, Daniela Buonanno, Danilo Atripaldi, Roberta Pepe, Giulia D’Alvano, Antonella Mangione, Simona Bonavita, Gabriella Santangelo, Alessandro Iavarone, Mario Cirillo, Fabrizio Esposito, Sandro Sorbi, Gioacchino Tedeschi

**Affiliations:** 1First Division of Neurology, Università degli Studi della Campania Luigi Vanvitelli, 80138 Naples, Italy; sabrina.esposito1@unicampania.it (S.E.); manueladestefano@hotmail.it (M.d.S.); mimmoip88@gmail.com (D.I.); dario.ricciardi89@gmail.com (D.R.); daniela.buonanno@policliniconapoli.it (D.B.); dalvanogiulia@hotmail.it (G.D.); gioacchino.tedeschi@unicampania.it (G.T.); 2Department of Advanced Medical and Surgical Sciences, MRI Research Center SUN-FISM, Università degli Studi della Campania Luigi Vanvitelli, 80138 Naples, Italy; federica_dinardo@fastwebnet.it (F.D.N.); matsic@hotmail.it (M.S.); giusysimona@hotmail.com (G.C.); daniloatripaldi@hotmail.it (D.A.); roby.pepe92@gmail.com (R.P.); antonella.mangione@gmail.com (A.M.); simona.bonavita@unicampania.it (S.B.); mario.cirillo@unicampania.it (M.C.); fabrizio.esposito@unicampania.it (F.E.); 3Division of Human Anatomy, Laboratory of Morphology of Neuronal Networks & Systems Biology, Department of Mental and Physical Health and Preventive Medicine, University of Campania Luigi Vanvitelli, 80138 Naples, Italy; giovanni.cirillo@unicampania.it; 4Department of Psychology, University of Campania Luigi Vanvitelli, 81100 Caserta, Italy; gabriella.santangelo@unicampania.it; 5Neurological Unit, CTO Hospital, AORN Ospedali Dei Colli, 80131 Naples, Italy; alessandro.iavarone@ospedalideicolli.it; 6IRCCS Fondazione Don Carlo Gnocchi ONLUS, 50143 Florence, Italy; sandro.sorbi@unifi.it; 7Department of Neuroscience, Psychology, Drug Research and Child Health (NEUROFARBA), University of Florence, 50134 Florence, Italy

**Keywords:** mild cognitive impairment, rTMS, resting state functional MRI, brain networks

## Abstract

Repetitive transcranial magnetic stimulation (rTMS) is a noninvasive neuromodulation technique that is increasingly used as a nonpharmacological intervention against cognitive impairment in Alzheimer’s disease (AD) and other dementias. Although rTMS has been shown to modify cognitive performances and brain functional connectivity (FC) in many neurological and psychiatric diseases, there is still no evidence about the possible relationship between executive performances and resting-state brain FC following rTMS in patients with mild cognitive impairment (MCI). In this preliminary study, we aimed to evaluate the possible effects of rTMS of the bilateral dorsolateral prefrontal cortex (DLPFC) in 27 MCI patients randomly assigned to two groups: one group received high-frequency (10 Hz) rTMS (HF-rTMS) for four weeks (*n* = 11), and the other received sham stimulation (*n* = 16). Cognitive and psycho-behavior scores, based on the Repeatable Battery for the Assessment of Neuropsychological Status, Beck Depression Inventory-II, Beck Anxiety Inventory, Apathy Evaluation Scale, and brain FC, evaluated by independent component analysis of resting state functional MRI (RS-fMRI) networks, together with the assessment of regional atrophy measures, evaluated by whole-brain voxel-based morphometry (VBM), were measured at baseline, after five weeks, and six months after rTMS stimulation. Our results showed significantly increased semantic fluency (*p* = 0.026) and visuo-spatial (*p* = 0.014) performances and increased FC within the salience network (*p* ≤ 0.05, cluster-level corrected) at the short-term timepoint, and increased FC within the left fronto-parietal network (*p* ≤ 0.05, cluster-level corrected) at the long-term timepoint, in the treated group but not in the sham group. Conversely, regional atrophy measures did not show significant longitudinal changes between the two groups across six months. Our preliminary findings suggest that targeting DLPFC by rTMS application may lead to a significant long-term increase in FC in MCI patients in a RS network associated with executive functions, and this process might counteract the progressive cortical dysfunction affecting this domain.

## 1. Introduction

Mild cognitive impairment (MCI) refers to a condition of cognitive decline greater than expected in relation to a patient’s age and education, which can affect all domains (i.e., memory, language, attention, visuospatial functioning, and executive functions), although it is not severe enough to impact daily functioning [1,2,3]. It is viewed as an intermediate stage between normal aging and dementia and as the prodromal stage for a variety of dementing neurodegenerative disorders. Much evidence suggests that individuals with MCI, although independent in daily activities, experience modifications in their quality of life (QOL) [4,5]. Moreover, neuropsychiatric symptoms are very common in individuals with MCI, ranging from 35% to 85% [6], and, together with functional decline, may contribute to reducing QOL in MCI patients in comparison to older adults without cognitive impairment [7]. In addition, executive dysfunction, referring to the impairment of “higher-level” cognitive functions involved in the control and regulation of “lower-level” cognitive processes and goal-directed, future-oriented behaviors, is a crucial feature in disease progression of both MCI and AD [8] and is common even in amnestic (aMCI) [9,10,11,12,13,14]. Remarkably, a growing body of evidence has shown that AD with a dysexecutive clinical phenotype progresses more rapidly than AD with an amnestic phenotype [8], and aMCI with a worsening of executive function and functional status, but not of memory, was more likely to progress to AD [15,16]. Consequently, interventions targeting executive and functional symptoms in MCI have been suggested as the most effective in improving QOL in this condition [7].

A number of pharmacological strategies have been evaluated (or are under investigation) for the treatment of MCI. Alternatively, nonpharmacological interventions for MCI (e.g., cognitive, physiological, nutritional supplementation, noninvasive brain stimulation, and psychosocial therapeutics) might play an important role in slowing or preventing the transition from MCI to dementia by approaches relatively free of adverse effects [17].

Noninvasive brain stimulation (NIBS) techniques, in particular, have provided therapeutic effects in several neuropsychiatric disorders, related to bi-directional changes in cortical excitability, with the direction of change depending on the choice of stimulation protocol [18,19]. Among NIBS techniques, transcranial magnetic stimulation (TMS) is currently used for studying the physiology of the central nervous system [20] and modulates the excitability of the cerebral cortex. Long-lasting after-effects are essential for the therapeutic efficacy and likely reflect the long-term modulation of brain networks activity, promoting cortical reorganization as well as modulation of neural activity beyond the stimulation period [21,22,23]. TMS can induce the long-term modulation of cortical excitability if delivered repetitively. Repetitive TMS (rTMS) consists in the application of a train of TMS pulses of the same intensity and at a given frequency to a single target brain area. In this way, the after-effects outlast the period of stimulation in a frequency-dependent manner: low frequency (≤1 Hz) rTMS reduces cortical excitability, whereas high-frequency (5–20 Hz) rTMS does the opposite [24].

Different rTMS protocols have demonstrated improvements in cognitive functions: for example, excitatory effects of the intermittent theta burst stimulation (TBS) [25,26] of the left dorsolateral prefrontal cortex (DLPFC) improves association memory, as well as the clinical symptoms and cognitive performance of subjects with AD [27]. Moreover, high-frequency (HF-rTMS) and low-frequency rTMS (LF-rTMS), seeking to enhance cognitive function in AD and MCI, have been shown to be safe and promising [28,29]. 

DLPFC is the most commonly used cortical target for therapeutic application of rTMS in MCI or AD-type dementia, considering the crucial role of this area in cognitive functions early impaired in AD, such as attention, executive functions, and working memory [30]. In particular, HF-rTMS of the left or right DLPFC [31,32,33,34,35] and of both left and right DLPFC [36,37] have been performed in AD and MCI patients, revealing a significant improvement in memory performances [32,33,35] and language comprehension [31] with a significant decrease in apathy [34]. Moreover, Cui et al. [35] revealed that the benefit on neuropsychological performance, as measured with the Auditory Verbal Learning Test (AVLT) score, lasted for two months beyond the rTMS protocol administration.

With regard to the potential application of neuroimaging techniques to assess modifications of brain connectivity induced by rTMS, analyses of the resting state functional magnetic resonance imaging (RS-fMRI) data or registration of electroencephalography during single-pulse TMS have been used to investigate functional connectivity (FC) properties within targeted RS brain networks (RSNs) in healthy subjects [38,39,40] and in AD patients [41]. Moreover, evidence of brain FC changes after the administration of rTMS to key nodes of the Default Mode Network (DMN), such as the posterior cingulate cortex (PCC) and precuneus, was revealed to be associated with the improvement of the related cognitive performances, such as episodic memory, especially when targeting cortical-hippocampal connections, in both healthy subjects [38,39,40] and in early AD [42] and MCI patients [35]. Interestingly, rTMS may be useful to modulate FC also in preclinical stages of AD, as revealed in APOE ε4 nondemented carriers who displayed RS-fMRI changes indicative of the normalization of FC within DMN after DMN-targeted rTMS [43]. Among RSNs, DMN has been revealed as critically impaired in MCI and AD [42,44,45,46] in posterior key nodes of DMN, represented by PCC and precuneus. These areas, together with bilateral hippocampi, are substantially involved in episodic memory and self-referential thoughts, which are the cognitive functions early impaired by neurodegeneration in AD [47,48]. Thus, DMN disconnections have been mostly studied as potential targets for preventive interventions in AD, such as those using rTMS [49]. However, in addition to DMN disruption, an altered functional connectivity within and/or between different, other brain networks, such as the executive control, salience, dorsal attention, and sensory-motor networks, has also been revealed in MCI and AD [50,51,52]. Conversely, evidence of FC changes in RSNs beyond DMN after administration of rTMS is still lacking. In particular, findings on the effects of multiple sessions of HF-rTMS over DLPFC on executive functions are controversial [53,54] and did not include monitoring through RS-fMRI analyses [55].

In order to shed more light on the still-unclear effects of rTMS over bilateral DLPFC on brain FC changes in MCI, we performed a RS-fMRI study with Independent Component Analysis (ICA) to quantify brain FC changes in a cohort of 27 MCI patients, randomized in two groups (i.e., stimulated—MCI-TMS—and sham—MCI-C—groups), who underwent a four-week HF-rTMS protocol in comparison to its sham counterpart. We performed neuropsychological assessment and whole-brain RS-fMRI and voxel-based morphometry (VBM) analyses at baseline and after five weeks and six months to assess modifications of cognitive performances and of brain FC and gray matter (GM) volume in the two groups across time. We expected to find RS-fMRI patterns of FC changes useful to better characterize the response of brain FC to rTMS across time in MCI, especially with regard to brain circuits related to the executive domain. Conversely, we expected to show no VBM changes across time in the MCI groups in order to confirm that the RS-fMRI findings were not related to GM atrophy.

## 2. Materials and Methods

### 2.1. Case Selection and Study Procedures

Twenty-seven MCI patients (14 males, 13 females; mean age 67.85 ± 9.28) were consecutively recruited at the First Division of Neurology of the University of Campania “Luigi Vanvitelli” (Naples, Italy) from January 2018 to February 2020. Patients were required to meet the following criteria: the “core” criteria for MCI as defined by the National Institute on Aging-Alzheimer’s Association workgroups on diagnostic guidelines for Alzheimer’s disease [4]; Clinical Dementia Rating (CDR) = 0.5; age ≥ 40 at the onset of cognitive symptoms; ability to understand and to sign the informed consent. Exclusion criteria for all subjects were: medical illnesses or substance abuse, interfering with cognitive functioning; any (other) major systemic, psychiatric, or neurological diseases; other causes of brain damage, including widespread cerebrovascular disorders at MRI; contraindications for MRI and TMS, according to the Standard Questionnaire of “The Safety of TMS Consensus Group” [56]. 

Thirteen right-handed healthy control subjects (HC) (5 males, 8 females; mean age 66.77 ± 9.08), who were age-, sex- and education-matched with the enrolled MCI patients and had no comorbid neurological, psychiatric, or significant medical conditions, were enrolled among caregivers’ friends. They underwent Mini-Mental State Examination (scores ≥ 27), Mental Deterioration Battery [57], and CDR = 0, which excluded the objective evidence of cognitive and functional impairment.

The research was conducted according to the principles expressed in the Declaration of Helsinki. Ethics approval was obtained from the Ethics Committee of the University of Campania “Luigi Vanvitelli” (Prot. N. 241/2017). Written informed consent was obtained from each participant.

#### Neuropsychological Assessment

To assess the cognitive functioning of the study groups, we used the Repeatable Battery for the Assessment of Neuropsychological Status (RBANS) Form A, B, and C [58]. These Forms comprise 12 subtests indexing 5 different cognitive domains (attention, immediate memory, delayed memory, language, visuospatial/constructional). To assess the behavioral profile of the study groups, we employed Italian versions of the Beck Depression Inventory-II (BDI-II), the Beck Anxiety Inventory (BAI), and the Apathy Evaluation Scale (AES). The BDI-II [59] questionnaire contains 21 self-report items used in clinical and research settings to assess depressive symptoms. The BAI [59] is a questionnaire with 20 self-report items focused on somatic, behavioral, emotional, and cognitive symptoms of anxiety. The AES [60,61] is an 18 item, self-assessment questionnaire, used for research purposes to evaluate apathy through 4 subscales (cognitive, behavioral, emotional, other).

### 2.2. Study Design

The study was designed as a randomized, controlled, and double-blind (patient and neuropsychologist) clinical study. Patients meeting all the inclusion criteria, and none of the exclusion criteria, underwent a baseline (T0) neuropsychological examination using RBANS Form A, BDI-II, BAI, and AES; then, they were randomized (http://www.random.org, accessed on 1 March 2020) to the active or sham arm. After the 4-week rTMS (T1), and 6 months after the end of the stimulations (T2), they repeated the neuropsychological assessment. To minimize the learning effect, RBANS Forms B and C were used in T1 and T2. The HCs underwent the same cognitive evaluation as the patients, at the same timepoints (T0, T1, T2).

### 2.3. Statistical Analysis: Between-Groups Comparisons of Clinical and Neuropsychological Data 

We tested the study variables for normality using both the Kolmogorov–Smirnov (K-S) test and asymmetry (values between −1 and +1 were considered acceptable) [62]; the departure from the normality distribution of part of the variables oriented us toward a nonparametric statistical approach (data not shown).

To verify the equivalence between the RBANS Forms, we compared via the Wilcoxon signed-rank test (Z) the performances of the HC group on subtest scores obtained from the RBANS forms.

At T0, we used the Kruskal–Wallis test (H), Mann–Whitney test (U), or Pearson’s chi-squared test (χ^2^ test) for comparing the three study groups on demographics (i.e., age, education, and sex) and for contrasting the MCI-TMS and MCI-C for neuropsychiatric screening measures (i.e., Neuropsychiatric Inventory). Moreover, we explored the T0 differences on more comprehensive cognitive (i.e., RBANS subtests) and behavioral (i.e., BDI-II; BAI; AES) measures via H completed by U for post hoc comparisons. 

At T1, to test the effects of TMS on RBANS subtests and behavioral measures, we compared the three study groups by Quade’s rank analyses of covariance [63] using the T0 measures as covariates.

All multiple and pairwise post hoc comparisons were corrected for Benjamini–Hochberg procedures; a Benjamini–Hochberg-corrected *p*-value ≤ 0.05 was considered statistically significant [64]. All analyses were performed using the IBM Statistical Package for Social Science (SPSS) version 20 (Chicago, IL, USA).

### 2.4. rTMS Protocol

Participants were randomly assigned in a double-blind condition to receive either active or sham rTMS. All the participants had no experience of rTMS, so they did not know whether they were receiving real or sham rTMS.

For the sham group, we used a placebo coil, with a mechanical outline and sound level (click) identical to the active one, which delivered <5% of the magnetic output. 

HF-rTMS (10 Hz) was applied over the DLPFC through a standard figure-of-eight coil with mean loop diameters of 9 cm connected to a Magstim2 Rapid stimulator (The Magstim Company, Whitland, UK). The coil was mounted on an articulated arm and positioned tangentially to the skull. The stimulation intensity was 80% of the resting motor threshold (RMT), defined as the lowest single pulse intensity required to produce a motor -evoked potential (MEP) greater than 50 μV (peak-to-peak amplitude) on more than five out of ten trials from the contracted contralateral abductor pollicis brevis (APB) [20]. 

rTMS was applied for ten minutes over the DLPFC (Brodmann areas 8/9) at the point located approximately 5 cm in a parasagittal plane parallel to the point of maximum stimulation of the APB, with the lowest possible intensity in five of ten stimuli [32]. Subjects assigned to the active group received on the DLPFC, bilaterally, HF-rTMS (10 Hz) for 10 min (20 trains of stimuli, each train consisting of 100 pulses and lasting 10 s, with wait-interval of 25 s; 2000 pulses/day). Each rTMS session was delivered 5 times/week on separate days for 4 weeks. The temporal order of HF-rTMS presentation was randomized and counter-balanced across the patients and the control subjects, in order to confirm that the results were not due to the temporal order of task presentation. The stimulation of the two hemispheres was performed sequentially at an interval of ten minutes.

### 2.5. MRI Analysis

#### 2.5.1. Magnetic Resonance Imaging

MR images were acquired on a 3T scanner equipped with a 32-channel parallel head coil (General Electric Healthcare, Milwaukee, WI, USA). The imaging protocol, according to a previous MRI analysis [65], included: three-dimensional T1-weighted sagittal images (gradient-echo sequence Inversion Recovery-prepared Fast Spoiled Gradient Recalled-echo, time repetition = 6.988 ms, TI = 650 ms, TE = 3.0 ms, flip angle = 9°, voxel size = 1 × 1 × 1 mm^3^; acquisition time = about 7 min) [66]; RS-fMRI was performed with a gradient-echo echo-planar imaging (GRE-EPI) sequence generating 320 T2*-weighted volumes of 44 axial slices (time repetition = 1500 ms, echo time = 19 ms, FA = 90°, voxel size = 3 × 3 × 3 mm^3^, matrix = 96 × 96, field of view = 288 mm, slice thickness = 3 mm, interslice gap = 0 mm; total acquisition time = ~8 min); T2-weighted fluid attenuation inversion recovery (FLAIR) was performed to exclude severe cerebrovascular disease according to standard clinical neuroradiological criteria on visual inspection by three experienced radiologists. During the functional scan, subjects were asked to simply stay motionless, awake, and relax and to keep their eyes closed. No visual or auditory stimuli were presented at any time during functional scanning. The total duration of each scan was about 38 min.

#### 2.5.2. RS-fMRI Data Preparation and Preprocessing 

Standard functional image data preparation and preprocessing, statistical analysis, and visualization were performed with the software BrainVoyager QX (Brain Innovation BV, Maastricht, The Netherlands). According to Friston et al. [67], data were processed by applying the correction for slice scan timing acquisition, a three-dimensional rigid-body motion correction (by a 6-parameter rigid body alignment to correct for minor head movements), a temporal high-pass filter with the cut-off set to 0.008 Hz, and a spatial smoothing of image series with a 6 mm full-width at half-maximum isotropic Gaussian kernel. Structural and functional data were coregistered and spatially normalized to the Talairach standard space using a 12-parameter affine transformation. To reduce the residual effects of head motion, as well as the effects of respiratory and cardiac signals, second-order motion and physiological nuisance correction were performed on the resulting image time series using a linear regression approach: the regression model included 24 motion-related predictors [67], with 6 head motion parameter time-series, their first-order derivatives, and the 12 corresponding squared parameter time-series; the mean time-courses from a white matter mask and a cerebrospinal fluid mask (as obtained from 3D-T1w spatial segmentation) were added as two additional predictors. In order to account for residual motion-related spikes, an additional spike-related regressor was created from the framewise displacement time-series, i.e., a predictor with a value of 1 at the time points of each detected spike and a value of 0 elsewhere [68,69]. To minimize the potential effects of head motion and possibly exclude subjects exhibiting excessive amounts of motion, we applied the following inclusion criteria: the six estimated head motion parameters (3 translation and 3 rotation) were considered and subjects exhibiting head translations >3 mm and/or head rotations >3 degrees were excluded from subsequent analyses. Then, the mean framewise displacement value (FD) was estimated as an additional measure of total instantaneous head motion [70,71], and the percentage of spike-corrupted volumes in each time-series was calculated. Potential spike-corrupted volumes were identified where the FD value exceeded a threshold of 0.5 mm; at this stage, subjects for whom the percentage of corrupted volumes exceeded 50% in the scan were also excluded from the analyses. 

#### 2.5.3. Resting State Network (RSN) Functional Connectivity Analysis

To extract RSN maps, single-subject and group-level independent component analyses (ICA) were carried out on the preprocessed functional time series using 2 plug-in extensions of BrainVoyager QX [72], respectively, implementing the fastICA algorithm [73] and the self-organizing group ICA algorithm [74], according to a previous MRI analysis [66]. Furthermore, an ICASSO step was added for the extraction of ICA components [66,75]. 

For each single subject, 50 independent components were extracted (corresponding to 1/6th of the number of time points) [76] and scaled to spatial z scores (i.e., the number of standard deviations of their whole-brain spatial distribution). According to Smith et al. [77], all individual component maps from all subjects were “clustered” in the subject space according to the mutual similarities of their whole-brain distributions using the self-organizing group ICA algorithm. Therefore, all 50 individual independent components were uniquely assigned to 1 out of 50 “clusters” of independent components. Once the components belonging to a cluster were selected, the corresponding maps were averaged and the resulting group map was taken as the representative FC pattern of the cluster. The 50 single-group average maps were visually inspected to recognize the spatial patterns associated with the main RSNs [77]. For this purpose, single-group 1-sample t tests were used to analyze the whole-brain distribution of the components in each group separately, and the resulting t maps were thresholded at *p* = 0.05 (Bonferroni-corrected over the entire brain) after regressing out age and gender from the series of individual maps at each voxel. An inclusive mask was finally created from the pooled healthy control and patient group baseline maps and used to define the search volume for within-network two-group comparisons. These comparisons were performed by fitting a two-way analysis of variance (ANOVA) model that included one between-subject factor with two levels (MCI-TMS and MCI-C) and one within-subject factor with three levels corresponding to the three time points of the study. From the fits of this model, post hoc t statistic contrasts were calculated for obtaining between-group t maps. The HC group was excluded from these comparisons because only patients had repeated time points; however, the HC group was contrasted with the entire group of MCI at baseline.

To correct the resulting t maps for multiple comparisons, regional effects within the search volume were only considered significant for compact clusters emerging from the joint application of a voxel- and a cluster-level threshold. The cluster-level threshold was estimated nonparametrically with a randomization approach: we calculated the FWHM from each RSN t map for the HC group and then, starting from an initial (uncorrected) threshold of *p* = 0.001 applied to all voxels [78], a minimum cluster size was calculated that protected against false-positive clusters at 5% after 1000 Monte Carlo simulations [79].

#### 2.5.4. Regional Atrophy Measurements: Voxel-Based Morphometry (VBM)

Structural MRI data analysis was performed using SPM12 (Statistical Parametric Mapping, Welcome Trust Centre for Neuroimaging, http://www.fil.ion.ucl.ac.uk/spm, accessed on 1 January 2021). The standard SPM pipeline for VBM analysis [80] was adapted only for longitudinal data: for each patient, the scans from all three time points were registered and bias-corrected using the serial longitudinal registration in SPM12. Thus, the output consisted of an average T1 image and three deformation fields. All time point images and average T1 were segmented into GM, white matter (WM), and cerebrospinal fluid (CSF). Spatial normalization was achieved by applying the high-dimensional DARTEL approach [81] using the GM and WM segments of the average T1 images. Each patient’s segmented GM time points were registered to their average using deformation fields obtained from serial registration, normalized in MNI space and smoothed with a Gaussian kernel of 8 mm full-width at half-maximum (FWHM). The flexible factorial model was specified, including the two patient groups and the three time conditions as two factors, while age, gender, and total intracranial volume were included as covariates of no interest. Statistical inference was performed at the voxel level, with a family-wise error (FWE) correction for multiple comparisons (*p* ≤ 0.05).

## 3. Results

### 3.1. Clinical and Neuropsychological Assessment

Out of 47 screened subjects, 7 did not fulfill the inclusion criteria for MCI or HC. Among the 40 subjects left, 13 were HCs and 27 were classified as MCI patients. A total of 11 MCI were randomly assigned to the active group and 16 to the sham group. All participants and progressive dropouts are reported in Figure 1.

As for the comparison between the RBANS Form A and B subtests, Semantic Fluency (Mdn: 19.00 vs. 13.00; Z = −2.97), Story Memory-IR (Mdn: 20.00 vs. 17.00; Z = −2.53), and Story Recall-DR (Mdn: 10.00 vs. 8.00; Z = −2.57) were not strictly equivalent as the HC group scored higher in Form A than in Form B (Benjamini–Hochberg-corrected *p*-value ≤ 0.05). At T0, we did not find statistically significant differences in demographics among the three study groups, and in neuropsychiatric screening measures between MCI-TMS and MCI-C (Table 1). 

A more comprehensive cognitive and behavioral assessment showed that the HC group scored higher than MCI-TMS or MCI-C on List Learning, Story Memory-IR, Semantic Fluency, Coding, List Recall, List Recognition, Story Recall-DR, and Figure Recall subtests of RBANS. Moreover, the MCI-TMS group obtained worse performance than MCI-C on the Semantic Fluency subtest. Finally, the MCI-TMS group had more marked apathetic symptoms than HC, but we did not find behavioral differences between MCI-TMS and MCI-C (Figure 2 and Figure 3; for numerical details, see Appendix A).

No significant side effects were reported in the MCI-TMS group after the four-week rTMS. Among the T1 evaluations, only the neuropsychological evaluation in T1 was available for the majority of enrolled subjects (23 MCI, 12 HC) and included in the analysis. Regarding the longitudinal neuropsychological and RS-fMRI assessment, we did not screen HC subjects versus MCI-TMS and MCI-C groups by repeated RS-fMRI exams across time. Moreover, the T2 examination (i.e., after six-months), including both neuropsychological and MRI assessments, was not collected in 14 subjects (1 MCI-TMS, 5 MCI-C, 8 HC) (Figure 1), as they were lost to follow-up. Taking into account these missing data, mostly regarding HC, the T2 neuropsychological assessment was not considered for the analysis. After five-weeks from baseline (T1), the MCI-TMS and HC groups scored higher than MCI-C on line orientation (*p* = 0.014) and semantic fluency (*p* = 0.026) subtests of RBANS B. Moreover, unlike the baseline, the MCI-TMS group did not show more apathetic symptoms when compared to MCI-C and HCs (Figure 2 and Figure 3; for numerical details, see Appendix A). 

### 3.2. Baseline RSN Functional Connectivity Analysis

The main RSN components were identified within the set of estimated independent components from the single-group maps using methodologies similar to previous studies (see, e.g., [82,83]). Among RSNs, within the cognitive domain, DMN, central executive network (CEN), right and left frontoparietal networks (FPNs), and salience network (SLN) components exhibited statistically significant regional group effects in their spatial distribution. In particular: the DMN comprised PCC, precuneus cortex, medial prefrontal cortex, and angular gyri [84]; the FPN was found as lateralized in a left and right network (L- and R-FPN), as also shown in previous studies [82,85,86]; the CEN included the anterior cingulate cortex [87], whereas the SLN encompassed the dorsolateral prefrontal, anterior insular, and inferior parietal cortices [83,88,89]. 

When compared to HCs, patients with MCI at baseline exhibited a decreased FC in the left angular gyrus within the DMN, in the superior parietal lobule within the R-FPN, and in the right medial frontal gyrus within the CEN. Conversely, when the two randomized patient groups were compared, retrospectively, between each other, no significant differences of FC were observed at baseline in the studied RSNs.

### 3.3. Five-Week and Six-Month RSN Functional Connectivity Analysis

With regard to MCI-C (*n* = 11), within-group comparisons did not show significant differences in FC in the studied RSNs after five weeks and after six months from baseline. Conversely, longitudinal analysis of RSN FC changes in MCI-TMS group (*n* = 10) revealed significant longitudinal effects across the three timepoints within the L-FPN and the SLN (Figure 4 and Figure 5). 

Within the L-FPN, within-group comparisons revealed a slight initial reduction in the FC at T1 (five week) vs. T0 (baseline), which was followed by a significant increase in the FC at T2 (six month) in the supramarginal gyrus (T2 vs. T0, T2 vs. T1), in the prefrontal cortex (T2 vs. T1), and in the middle frontal gyrus (T2 vs. T0, T2 vs. T1) (cluster-level-corrected *p* ≤ 0.05, voxel-level *p* ≤ 0.001) (Figure 4). 

Within the SLN, within-group comparisons revealed an increased FC in the left parahippocampal gyrus and in the left superior temporal gyrus at T1 vs. T0 (cluster-level corrected *p* ≤ 0.05, voxel-level *p* < 0.001), which was not present at T2 vs. T0 (Figure 5).

### 3.4. VBM Analysis

The applied flexible factorial model showed that the MCI-TMS group was significantly more atrophic than the MCI-C group in a single cluster in the right superior frontal gyrus (MNI coordinates: 24; 34; 41). However, no significant differences were observed between the two groups across the three timepoints.

## 4. Discussion

Even if our intervention was of short duration and the sample size reduced across time, the present preliminary study suggests that noninvasive brain stimulation may improve cognitive performances in MCI with potential effects on brain FC extended up to six months after treatment. In particular, the short-term observation, after five weeks from rTMS (T1), showed better performances in semantic fluency and orientation line tests and increased FC in the left para-hippocampal and superior temporal gyri within the SLN of the MCI-TMS group in comparison to nonstimulated MCI patients. The long-term observation, after six months from rTMS (T2), showed increased FC in the left supramarginal and middle frontal gyri and in the left prefrontal cortex within the L-FPN. Our preliminary findings, which combine for the first time neuropsychological and RS-fMRI data for monitoring short-term and long-term effects of rTMS in MCI, may corroborate the hypothesis that treatments based on multiple sessions of rTMS may represent a promising tool for influencing cognition in persons with neurodegenerative diseases [90].

We revealed that, in the short post-treatment period, rTMS intervention exhibited a broad modulation effect on cognitive functions and brain FC in MCI patients. Potential effects of our rTMS protocol were observed after five weeks with an improvement of executive (i.e., semantic fluency) and visuo-spatial (i.e., line orientation) performances and with an increase in FC of the left para-hippocampal and superior temporal gyri within the SLN (i.e., comprising brain regions involved in memory and language domains, as well as in the processing of visual and auditory stimuli) in the MCI-TMS group compared to MCI-C. Moreover, apathetic symptoms, reported at baseline in the MCI-TMS group in comparison to HCs and MCI-C groups, were not detected immediately after treatment, suggesting a trend toward a reduction in behavioral impairment in the treated group. Similar immediate broad effects on cognitive functions after rTMS of the left DLPFC have been reported in MCI patients with comorbid Parkinson’s disease treated by the 10-session iTBS protocol and monitored by neuropsychological assessment and 99mTc-TRODAT SPECT [55]. However, evidence regarding the potential effects of rTMS on executive functions are conflicting: some authors found mild significant effects of improvements in executive functions (i.e., Trail Making Test and Stroop test performances) after rTMS in patients with treatment-resistant major depression [91,92,93,94,95]; conversely, other studies, focused on the effects of rTMS on cognitive functions in MCI and AD [32,35,96], did not report significant differences in performances in the executive domain by comparing treated patients and control groups. To note, the wide variety of cognitive measurement tools across studies, due to the evidence that executive function is a complex and multi-dimensional cognitive construct [97], may have contributed to the variation in research results. Moreover, heterogeneity in rTMS protocols could have influenced the described effects on cognitive functions. The stimulation parameters that have been shown more associated with cognitive improvement in MCI patients include high frequency (5–20 Hz) rTMS (especially of bilateral DLPFC or prefrontal cortex) for >20 treatment sessions [54]. With regard to the reduction in apathetic symptoms in MCI-TMS patients immediately after treatment, there is some evidence that HF-rTMS (especially at 10 Hz) seems to be a promising technique to improve negative symptoms in patients with schizophrenia [98,99,100], thereby recalling the trend toward apathy reduction observed in our MCI-TMS group.

The long-term effects of rTMS treatment, after six months from baseline, although impacted by the dropouts of MCI patients across time, suggested potential effects on the FC in L-FPN in some regions crucial for executive function: left supramarginal and middle frontal gyri and left prefrontal cortex. In particular, those regions, together with other left fronto-temporo-parietal areas, are anatomically related to verbal fluency, as recently underlined by a disconnectome study performed in patients with acute ischemic stroke [101], as well as revealed in other lesion-symptom mapping studies in patients with ischemic stroke or brain tumors [102,103,104,105]. With regard to verbal fluency components, Biesbroek et al. [101] described overlapping correlates for both phonemic and semantic fluency in connections from the left thalamus to the prefrontal cortex, and, to a lesser degree, in widespread left hemispheric frontoparietal and temporal regions. A more extensive left frontoparietal/peri-sylvian network was associated specifically to phonemic fluency, while several clusters of left temporal voxels and clusters in thalamic nuclei that project to the temporal lobe were specifically related to semantic fluency [101]. These results recalled the theory that phonemic fluency depends more on left frontoparietal regions and semantic fluency depends more on left temporal regions [104], according to the motor-phonological and lexical-semantic streams underlying the two models of speech processing. Additionally, Biesbroek et al. [101] revealed a dissociation within the left temporal lobe, with posterior and medial temporal regions being involved in semantic fluency, and anterolateral temporal regions in phonemic fluency. Disconnectome analysis by Biesbroek et al. [101] showed a specific involvement of a larger extent of peri-sylvian and frontoparietal white matter in phonemic fluency, probably because of the stronger dependency of phonemic fluency on the so-called phonological loop that involves the superior temporal gyrus, supramarginal gyrus, and inferior frontal regions compared to semantic fluency [106]. In the light of this evidence, our findings of increased FC in L-FPN in the left supramarginal and middle frontal gyri, interpreted as long-term effects of rTMS of bilateral DLPFC in MCI patients, may suggest a potential long-lasting improving effect of rTMS on verbal fluency function, as predictively revealed by the RBANS scores collected after the short-term neuropsychological assessment. However, the lack of neuropsychological data after six months from baseline hinders the clinical support of the hypothesis of a potential long-term benefit of rTMS of DLPFC on verbal fluency in our MCI patients.

We did not find significant differences in whole-brain GM atrophy across time in the two studied groups. However, the short-/long-term potentiation of synapses and rapid dynamic alterations in GM density have been previously reported after rTMS treatment [107,108], suggesting that structural changes could be triggered by high-frequency rTMS pulses. The occurrence of structural alterations has been related to the triggering of structural neuroplasticity as a counterpart of changes in functional processing, especially with regard to the effects of rTMS on strengthening brain WM connectivity between the left DLPFC and insula targeted by rTMS stimulation [109]. Conversely, the time-course of GM changes in the study of May et al. [108], after a 5-day treatment with 1 Hz rTMS at an intensity of 110% motor threshold (MT), suggested that GM modification could reflect the fast adjustment of neuronal systems at the cellular level, such as spine and synapse turnover, rather than slower evolving mechanisms, such as neuronal or glial cell genesis [18,110]. Probably, this last evidence could explain the lack of significant long-lasting changes in GM volume after six months from baseline in both treated and untreated patients, as resulted from our VBM analysis. 

Our preliminary study was limited in several aspects, and these limitations suggest important insights for future research. First, the sample size was small. Only 27 patients were included in the analysis and 14 subjects were lost at T2: this could have hindered the revealing of significant effects on RS-fMRI changes due to the reduction in observations across time, thereby inducing a lack of power or high variability. Second, we did not screen HC subjects versus active rTMS and sham control groups by RS-fMRI across time, which would have confirmed that the RS-fMRI features could reflect the clinical efficacy of rTMS in MCI across time. Third, we did not collect cognitive and behavioral data after six months from rTMS stimulation, which would have reinforced the hypothesis of long-lasting effects of rTMS of DLPFC on verbal fluency and apathy in MCI patients.

## 5. Conclusions

Our findings revealed that, in the short post-treatment period, rTMS intervention may induce a broad modulation of cognitive functions (i.e., verbal fluency and line orientation) and brain FC (i.e., in SLN) in MCI patients, showing, after six months, long-term effects on FC in L-FPN in two regions known to be crucial for executive functioning (i.e., left supramarginal and middle frontal gyri). These results may support the development of noninvasive interventions for persons at risk of dementia, particularly due to AD, in the longer term. It would be of great interest to establish in future large-scale studies whether the observed effects can be enhanced and transformed into longer-lasting and clinically relevant changes by means of rTMS sessions and by combining TMS with cognitive rehabilitation in MCI patients.

## Figures and Tables

**Figure 1 biomedicines-10-00994-f001:**
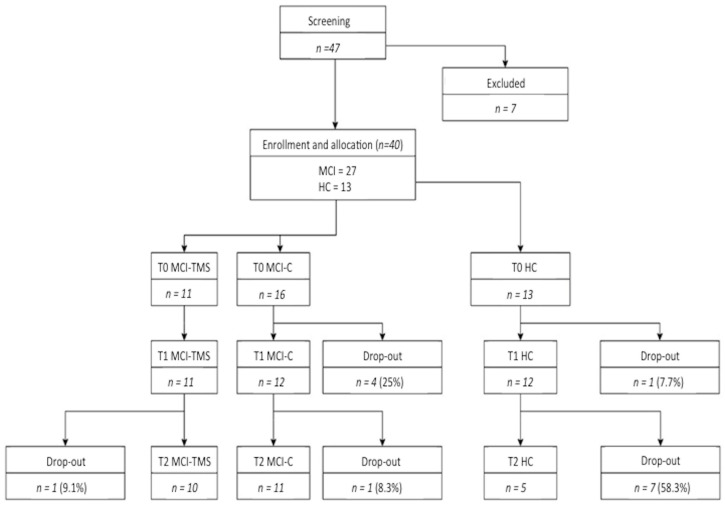
Flow diagram of referred and enrolled patients.

**Figure 2 biomedicines-10-00994-f002:**
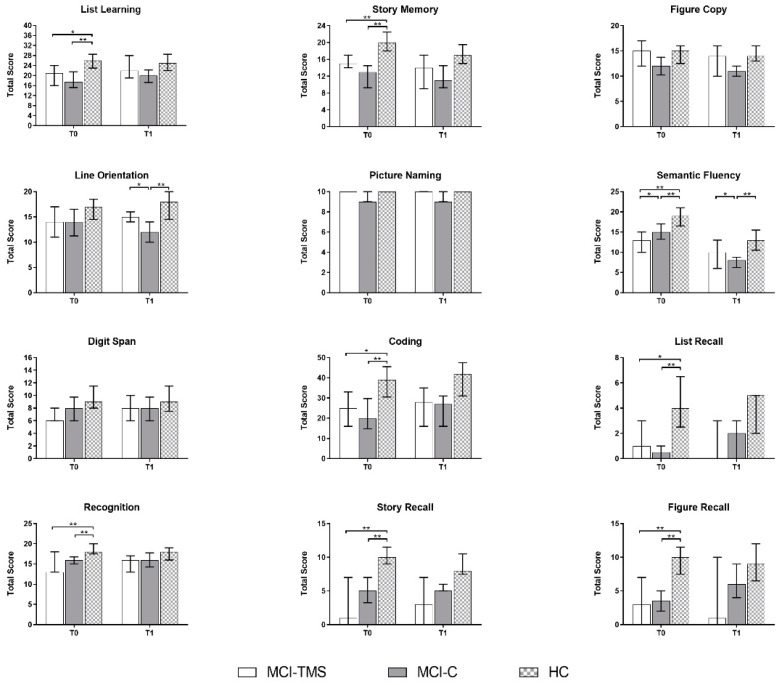
Between-group comparison on Repeatable Battery for the Assessment of Neuropsychological Status (RBANS) sub-tests at T0 and T1 (pre- and post-treatment) using Quade’s rank analyses of covariance; Benjamini–Hochberg-corrected * *p* ≤ 0.05, ** *p* ≤ 0.01.

**Figure 3 biomedicines-10-00994-f003:**
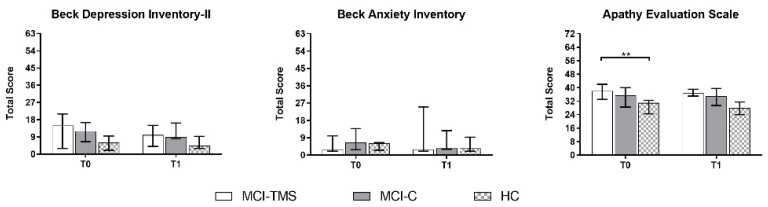
Between-group comparison on behavioral measures at T0 and T1 (pre- and post-treatment) using Quade’s rank analyses of covariance; Benjamini–Hochberg-corrected ** *p* ≤ 0.01.

**Figure 4 biomedicines-10-00994-f004:**
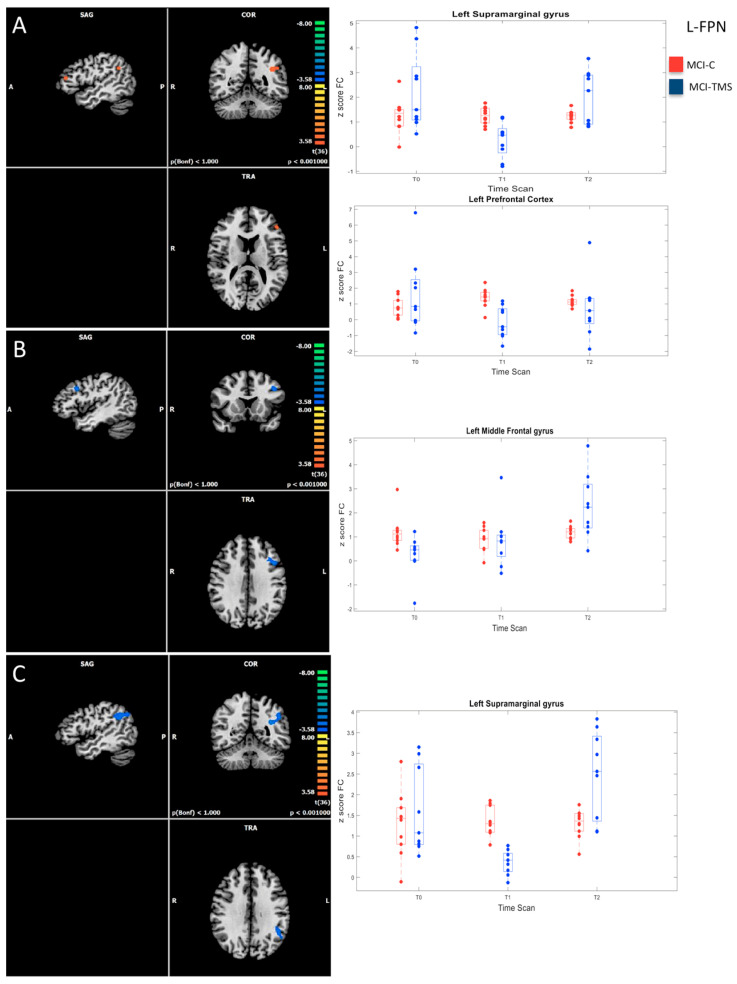
On the left: whole-brain significant connectivity differences between MCI−TMS patients at different time points (*p* ≤ 0.001 cluster-level-corrected) within left frontoparietal network. (**A**) T0 MCI−TMS vs. T1 MCI−TMS. (**B**) T0 MCI−TMS vs. T2 MCI−TMS. (**C**) T1 MCI−TMS vs. T2 MCI−TMS. On the right: corresponding box−plot of the average ICA z−scores. We also reported values extracted from the MCI−C group at different time points to show a relatively constant FC in these subjects who have not undergone stimulation. No significant difference was present for other comparisons.

**Figure 5 biomedicines-10-00994-f005:**
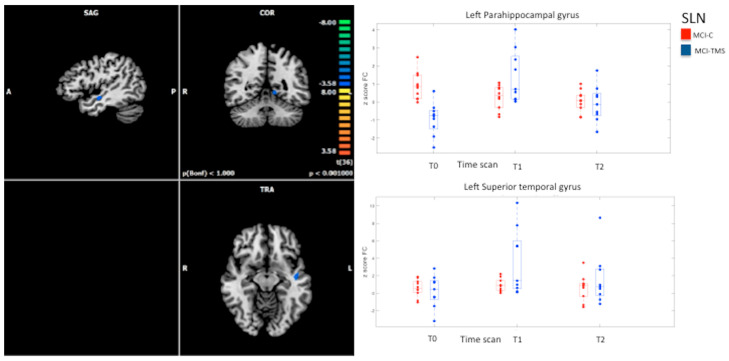
On the left: whole-brain significant connectivity differences between T0 MCI−TMS and T1 MCI−TMS patients (*p* ≤ 0.001 cluster-level-corrected) within salience network. On the right: corresponding box-plot of the average ICA z−scores. We also reported values extracted from the MCI−C group at different time points to show a relatively constant FC in these subjects who have not undergone stimulation. No significant difference was present for other comparisons.

**Table 1 biomedicines-10-00994-t001:** Between-group comparison on demographics and neuropsychiatric screening measures at pre-treatment evaluation; data are reported as median (25th, 75th percentile) or count (percentage).

Variable	MCI-TMS (*n* = 11)	MCI-C (*n* = 16)	HC (*n* = 13)	^a^ H-Test; ^b^ U-test; ^c^ χ^2^ Test	*p*-Value	Adj-p	MCI-TMS vs. MCI-C	MCI-TMS vs. HC	MCI-C vs. HC
Demographics									
Age, years	64.00 (60.00, 74.00)	70.50 (62.50, 77.25)	68.00 (60.50, 74.50)	^a^ 1.68	0.431	0.895	-	-	-
Education, years	13.00 (10.00, 13.00)	11.00 (8.00, 13.00)	13.00 (13.00, 18.00)	^a^ 6.06	0.048	0.544	-	-	-
Sex, male	6 (46.20%)	8 (50.00%)	5 (45.50%)	^c^ 0.06	0.967	1.000	-	-	-
Neuropsychiatric symptoms									
Neuropsychiatric Inventory dimensions									
Delusions	0.00 (0.00, 0.00)	0.00 (0.00, 0.00)	*	^b^ 71.50	0.375	0.895	-	-	-
Hallucination	0.00 (0.00, 0.00)	0.00 (0.00, 0.00)	*	^b^ 77.00	1.000	1.000	-	-	-
Agitation/aggression	0.00 (0.00, 0.00)	0.00 (0.00, 0.00)	*	^b^ 63.00	0.103	0.544	-	-	-
Dysphoria	6.00 (4.00, 12.00)	9.00 (0.00, 9.75)	*	^b^ 75.50	0.933	1.000	-	-	-
Anxiety	6.00 (0.00, 12.00)	9.00 (4.00, 12.00)	*	^b^ 61.00	0.370	0.895	-	-	-
Euphoria	0.00 (0.00, 0.00)	0.00 (0.00, 0.00)	*	^b^ 71.50	0.375	0.895	-	-	-
Apathy	4.00 (0.00, 9.00)	0.00 (0.00, 9.00)	*	^b^ 68.00	0.593	0.988	-	-	-
Disinhibition	0.00 (0.00, 0.00)	0.00 (0.00, 1.50)	*	^b^ 60.50	0.109	0.544	-	-	-
Irritability	4.00 (0.00, 9.00)	4.00 (0.00, 6.75)	*	^b^ 64.50	0.478	0.895	-	-	-
Aberrant motor activity	0.00 (0.00, 0.00)	0.00 (0.00, 0.00)	*	^b^ 77.00	1.000	1.000	-	-	-
Night-time behavioural disturbances	0.00 (0.00, 0.00)	0.00 (0.00, 0.00)	*	^b^ 77.00	1.000	1.000	-	-	-
Appetite and eating abnormalities	0.00 (0.00, 0.00)	0.00 (0.00, 0.00)	*	^b^ 77.00	1.000	1.000	-	-	-

Note. * Healthy control group was not tested; MCI-TMS, patient with Mild Cognitive Impairment and underwent TMS; MCI-C, patients with Mild Cognitive Impairment and did not undergo TMS; HC, healthy controls; Adj-p represents *p*-value corrected for multiple comparisons using Benjamini–Hochberg procedure; statistically significant differences are shown in bold. ^a, b, c^ label the Kruskal–Wallis test (H), Mann–Whitney test (U), or Pearson’s chi-squared test (χ^2^ test) for comparing the three study groups on demographics (i.e., age, education, and sex) and for contrasting the MCI-TMS and MCI-C for neuropsychiatric screening measures.

## Data Availability

All data and materials support the reported claims and comply with standards of data transparency. Data will be made available on reasonable request.

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
