# Peer review of "Repetitive Transcranial Magnetic Stimulation (rTMS) of Dorsolateral Prefrontal Cortex May Influence Semantic Fluency and Functional Connectivity in Fronto-Parietal Network in Mild Cognitive Impairment (MCI)"

_biomedicines, 2022, doi:10.3390/biomedicines10050994_

Round 1

Reviewer 1 Report

The present article reports a double-blind randomized study to investigate the efficacy of rTMS (applied to DLPFC) in patients with MCI, to ameliorate cognitive symptoms, and investigate changes in brain connectivity and structure.

The manuscript is very well written and all details are reported in a convincing manner. There are some intrinsic limitations in the missing data, but the authors describe their implications in the discussion. The interpretation are sound and in line with the evidence collected. I have only a few comments about the methods and analysis

Main comment 1)

It seems that there are some discrepancies in the analysis, associated with missing data. In particular, neuropsychological assessment has been conducted only for t0 and t1. Analysis for fMRI data has been conducted also for T2 (even if, as it is visible in figure 4, only with very few subjects): the analyses should be more consistent. If the drop-out/missing data are different for the type of data (e.g. neuropsychological tests or fMRI data), this should be clarified. In such cases, for completeness, the authors should report (in supplementary material) the number of observations for each type of data.

Es. p. 20 line 35. It is not clear if the missing data in patients are the same for clinical and for fMRI analysis. If this is the case (as it appears from dot plots in Figure 4).

Main comment 2)

There is no correlation analysis between functional and behavioral measures. This is of major importance for the interpretation, to understand whether the observed changes reflect the same or different underlying features.

Main comment 3)

In some cases of comparison with T2 data (where more missing data are present), the lack of significant effects (as those found for fMRI) could be related indeed to the difference in observations (par 3.3 line 35). This lack of difference could reflect a lack of power or high variability, rather than a real lack of differences. This should be better stressed in the discussion (that is that the few participants in the study and missing data could have more relevant effects for some results rather than others).

Other comments

  1. 5 Line 200, values -1 and +1 (what test was used here?)
  2. 5 Line 237 were the 5 sessions in separate days? Please specify here.

Figure 2, and Figure 3. The figure caption should report the type of analysis used for the comparison. This is particularly important as (correctly) different analyses. were used for T0 and T1.

p.21 about the exclusion of Healthy control in fMRI analysis, this detail could be rehearsed in the result section. The manuscript is quite long, and a reader could miss this important aspect, and wonder why HCs were not included.

  1. 22 it seems that HCs were not included in VBM analyses, please clarify.

Please note that due to a section break for Table 1, both page numbers and line numbers started again from the beginning in the manuscript.

With Best Regards

Author Response

Reviewer#1

The present article reports a double-blind randomized study to investigate the efficacy of rTMS (applied to DLPFC) in patients with MCI, to ameliorate cognitive symptoms, and investigate changes in brain connectivity and structure.

The manuscript is very well written and all details are reported in a convincing manner. There are some intrinsic limitations in the missing data, but the authors describe their implications in the discussion. The interpretation are sound and in line with the evidence collected. I have only a few comments about the methods and analysis

Authors: We thank the Reviewer#1 for her/his encouraging comments.

Main comment 1)

It seems that there are some discrepancies in the analysis, associated with missing data. In particular, neuropsychological assessment has been conducted only for t0 and t1. Analysis for fMRI data has been conducted also for T2 (even if, as it is visible in figure 4, only with very few subjects): the analyses should be more consistent. If the drop-out/missing data are different for the type of data (e.g. neuropsychological tests or fMRI data), this should be clarified. In such cases, for completeness, the authors should report (in supplementary material) the number of observations for each type of data.

 Authors: We thank the Reviewer#1 for having suggested to clarify this crucial point. We checked and confirmed that the analysis of data was performed in the subjects sample as displayed in the flow-chart (Fig.1). In particular, we underlined in the results that “Regarding the longitudinal neuropsychological and RS-fMRI assessment, we did not screen HC subjects versus MCI-TMS and MCI-C groups by repeated RS-fMRI exams across time. Moreover, the T2 examination (i.e., after six-months), including both neuropsychological and MRI assessments, was not collected in 14 subjects (1 MCI-TMS, 5 MCI-C, 8 HC) (Fig.1), as they were lost to follow-up. Taking into account these missing data, mostly regarding HC, the T2 neuropsychological assessment was not considered for the analysis” (p.8, lines 381-388).

Es. p. 20 line 35. It is not clear if the missing data in patients are the same for clinical and for fMRI analysis. If this is the case (as it appears from dot plots in Figure 4).

 Authors: According to the appropriate request of clarification from the Reviewer#1, we specified that the missing data were the same for clinical and fMRI analysis, regarding MCI-TMS and MCI-C groups (p.8, lines 381-388). Moreover, we underlined that we did not screen HC subjects versus MCI-TMS and MCI-C groups by RS-fMRI across time (p.8 lines 381-383; p.14, lines 558).

Main comment 2)

There is no correlation analysis between functional and behavioral measures. This is of major importance for the interpretation, to understand whether the observed changes reflect the same or different underlying features.

Authors: We thank the Reviewer#1 for this valuable observation. However, we did not include T2 neuropsychological and RS-fMRI data due to missing data, thereby hindering to correlate functional and behavioral measures across the six months of observation. We underlined this crucial point in the limits section: “we did not collect cognitive and behavioral data after six months from rTMS stimulation, which would have reinforced the hypothesis of long-lasting effects of rTMS of DLPFC on verbal fluency and apathy in MCI patients” (p. 14, lines 560, 563).

Main comment 3)

In some cases of comparison with T2 data (where more missing data are present), the lack of significant effects (as those found for fMRI) could be related indeed to the difference in observations (par 3.3 line 35). This lack of difference could reflect a lack of power or high variability, rather than a real lack of differences. This should be better stressed in the discussion (that is that the few participants in the study and missing data could have more relevant effects for some results rather than others).

Authors: We thank the Reviewer#1 for this valuable observation which allows to better discuss our findings. We underlined throughout the text and mainly in the discussion the impact of the missing data on the obtained results (p.11, lines 426, 428; p.13, line 506-507; p 14, lines 554-557).

Other comments

  1. 5 Line 200, values -1 and +1 (what test was used here?)

Authors: We specified that we tested the study variables for normality using both the Kolmogorov-Smirnov (K-S) test and asymmetry (values between -1 and +1 were considered acceptable) (62) (p. 4, line 203).

  1. 5 Line 237 were the 5 sessions in separate days? Please specify here.

Authors: We specified that the five sessions took place in separate days (p. 5, line 244).

Figure 2, and Figure 3. The figure caption should report the type of analysis used for the comparison. This is particularly important as (correctly) different analyses were used for T0 and T1.

Authors: As suggested, we reported in the figures (2-3) captions the type of analysis used (i.e., Quade's rank analyses of covariance; Benjamini–Hochberg corrected *p<0.05, ** p<0.01).

p.21 about the exclusion of Healthy control in fMRI analysis, this detail could be rehearsed in the result section. The manuscript is quite long, and a reader could miss this important aspect, and wonder why HCs were not included.

Authors: According to this valuable comment, we underlined in the results that HC did not perform RS-fMRI examination across time (p. 8, line 381-383).

  1. 22 it seems that HCs were not included in VBM analyses, please clarify.

Authors: According to a suggestion of Reviewer#2, we clarified that the comparison between MCI and HC subjects was not performed, having  the objective of exploring only potential VBM changes across time in the MCI groups (in order to have the GM atrophy data together with RS-fMRI data) (p. 4, line 151-153, p.7, line 343).

Please note that due to a section break for Table 1, both page numbers and line numbers started again from the beginning in the manuscript.

Authors: We paid more attention to this editing mistake in the revised version of the manuscript. We checked also the revised .pdf file in this regard.

With Best Regards

Reviewer 2 Report

This is a well-written, well-executed and interesting report on the effect of repetitive transcranial magnetic stimulation of dorsolateral prefrontal cortex in MCI. The strengths of the paper include comprehensive neuropsychological testing, accompanying ICA rs-fMRI, and complementary longitudinal structural imaging. The paper is likely to generate considerable interest.  

Minor suggestions

1, As this is a relatively small study attributing therapeutic benefits to an intervention, the authors rightly adopt a careful stance. As this is not a formal clinical trial, I would emphasise in the text that this is a “pilot study”, with promising “preliminary” findings which need replication in much larger cohorts. For example, I would probably tone down the last sentence of the abstract to “our preliminary finding etc…”. I am wondering if the title should be cautiously adjusted to …rTMS of DLPFC “may” influence… 

2, Ethics approval is clearly stated in the manuscript, but in other jurisdictions this type of study with a putative therapeutic intervention, may need to be registered as a clinical trial. It may be worth stating that this was not deemed necessary by the authors or the ethics committee if that it is the case.

3, Was a pooled MCI (both treated and untreated, n=27) versus healthy control VBM contrast considered? If this is not available, it is not necessary to perform it as it has limited relevance to the effect of rTMS. However, if these findings are available, it may be interesting to present whether parietal /mesial temporal changes may have been detected.

4, This reviewer wouldn’t attach too much importance to the absence of VBM effect. What was the authors’ hypothesis with regards to VBM? Resolution of atrophic changes in the treatment arm (HF-rTMS n = 11) would be a quixotic expectation and slower progression in GM loss is unlikely to be detected in six months. It may be interesting to state the hypotheses at the end of the introduction. 

5, Very minor remarks; the abbreviations are not entirely consistent throughout the text; under section 3.4 VBM analysis the acronym “TMS-MCI” is used whereas everywhere else it is “MCI-TMS”. Also in the last sentence of the abstract “..in an RS network …” should probably be amended to ““..in a RS network …”

Author Response

Reviewer#2

This is a well-written, well-executed and interesting report on the effect of repetitive transcranial magnetic stimulation of dorsolateral prefrontal cortex in MCI. The strengths of the paper include comprehensive neuropsychological testing, accompanying ICA rs-fMRI, and complementary longitudinal structural imaging. The paper is likely to generate considerable interest.  

Authors: We thank the Reviewer#2 for her/his encouraging comments.

Minor suggestions

1, As this is a relatively small study attributing therapeutic benefits to an intervention, the authors rightly adopt a careful stance. As this is not a formal clinical trial, I would emphasise in the text that this is a “pilot study”, with promising “preliminary” findings which need replication in much larger cohorts. For example, I would probably tone down the last sentence of the abstract to “our preliminary finding etc…”. I am wondering if the title should be cautiously adjusted to …rTMS of DLPFC “may” influence… 

Authors: We agree with this valuable suggestion of the Reviewer#2 and toned down some sentences regarding the generalizability/impact of our findings in the title (“Repetitive transcranial magnetic stimulation (rTMS) of dorso-lateral prefrontal cortex may influence semantic fluency and functional connectivity in fronto-parietal network in Mild Cognitive Impairment (MCI)”) and throughout the text (pp. 1-2, 12-14).

2, Ethics approval is clearly stated in the manuscript, but in other jurisdictions this type of study with a putative therapeutic intervention, may need to be registered as a clinical trial. It may be worth stating that this was not deemed necessary by the authors or the ethics committee if that it is the case.

Authors: We thank for this appropriate observation regarding the type of the study presented, using a therapeutic intervention. We did not register this study as a clinical trial, in agreement with the Ethics committee of our institution, because rTMS is widely approved in Italy for rehabilitative purposes and, then, we could have treated our MCI patients in this interventional, longitudinal and controlled study by outpatient. In this regard, we modified the term “clinical trial” in “clinical study” in the study design (p. 4, line 194).

3, Was a pooled MCI (both treated and untreated, n=27) versus healthy control VBM contrast considered? If this is not available, it is not necessary to perform it as it has limited relevance to the effect of rTMS. However, if these findings are available, it may be interesting to present whether parietal /mesial temporal changes may have been detected.

Author: Also according to a comment by Reviewer#1, we specified that the comparison between MCI and HC subjects was not performed, having  the objective of exploring only potential VBM changes across time in the MCI groups (in order to have the GM atrophy data together with RS-fMRI data) (p. 4, line 151-153; p.7, line 343).

  1. This reviewer wouldn’t attach too much importance to the absence of VBM effect. What was the authors’ hypothesis with regards to VBM? Resolution of atrophic changes in the treatment arm (HF-rTMS n = 11) would be a quixotic expectation and slower progression in GM loss is unlikely to be detected in six months. It may be interesting to state the hypotheses at the end of the introduction. 

Authors: We thank the Reviewer#2 for her/his intriguing observation regarding the absence of VBM effect across time. Stimulated to express the hypothesis regarding the monitoring of atrophic changes across time by VBM analysis, we specified in the introduction that “we expected to show no VBM changes across time in the MCI groups in order to confirm that the RS-fMRI findings were not related to GM atrophy” (p. 4, lines 151-153).

5, Very minor remarks; the abbreviations are not entirely consistent throughout the text; under section 3.4 VBM analysis the acronym “TMS-MCI” is used whereas everywhere else it is “MCI-TMS”. Also in the last sentence of the abstract “..in an RS network …” should probably be amended to ““..in a RS network …”

Authors: We apologize for the identified inaccuracies and corrected them throughout the text.
